# Association of SOD2 (rs4880) and GPX1 (rs1050450) Gene Polymorphisms with Risk of Balkan Endemic Nephropathy and its Related Tumors

**DOI:** 10.3390/medicina55080435

**Published:** 2019-08-03

**Authors:** Biljana Dragicevic, Sonja Suvakov, Djurdja Jerotic, Zorica Reljic, Ljubica Djukanovic, Ivanka Zelen, Marija Pljesa-Ercegovac, Ana Savic-Radojevic, Tatjana Simic, Dejan Dragicevic, Marija Matic

**Affiliations:** 1Clinic for Otorhinolaryngology and Maxillofacial Surgery, Clinical Center of Serbia, Pasterova 2, 11000 Belgrade, Serbia; 2Institute of Medical and Clinical Biochemistry, Faculty of Medicine, University of Belgrade, 11000 Belgrade, Serbia; 3Faculty of Medicine, University of Belgrade, 11000 Belgrade, Serbia; 4Medical laboratory “PAN LAB”, 36000 Kraljevo, Serbia; 5Department of Biochemistry, Faculty of Medical Sciences, University of Kragujevac, 34000 Kragujevac, Serbia; 6Serbian Academy of Sciences and Arts, 11000 Belgrade, Serbia; 7Clinic of Urology, Clinical Centre of Serbia, Resavska 51, 11000 Belgrade, Serbia

**Keywords:** Balkan endemic nephropathy, superoxide dismutase, glutathione peroxidase, gene polymorphism

## Abstract

*Background and Objectives:* Experimental data show that superoxide dismutase 2 (SOD2) is involved in ochratoxin (OTA)-induced nephrotoxicity, whereas clinical data indicate the role of *SOD2* rs4880 or glutathione peroxidase 1 (*GPX1)* rs1050450 polymorphisms in end-stage renal disease and urothelial carcinoma risk, known to be the major complications of Balkan endemic nephropathy (BEN). Therefore, we hypothesized that *SOD2* and *GPX1* gene polymorphisms would influence the risk of BEN and its associated tumors. *Materials and*
*Methods:* The study was conducted in 207 BEN patients and 86 controls from endemic areas. *Results:* Individuals with both copies of variant *SOD2* allele, known for lower mitochondrial antioxidant protection, are at a significantly higher BEN risk (OR = 2.6, *p* = 0.021). No association was observed between *GPX1* gene polymorphism and BEN risk. Combining *SOD2* and *GPX1* genotypes did not alter the risk of BEN development. Regarding the risk of urothelial tumors in BEN patients, none of the polymorphisms studied was significantly associated with the risk of these tumors. *Conclusions:* Polymorphism in *SOD2* rs4880 gene affects the risk of BEN development. Hence, *SOD2* genotyping could, together with a panel of other enzymes, be used as a biomarker of susceptibility in BEN areas.

## 1. Introduction

Balkan endemic nephropathy (BEN) is a form of chronic tubulointerstitial nephritis which will eventually lead to end-stage renal disease (ESRD) [1]. From the first description of the disease, nearly 70 years ago, until nowadays, its geographic distribution has remained the same. Namely, this nephropathy affects individuals living in certain rural areas along of the Danube River and its tributaries in Serbia, Bosnia, Croatia, Bulgaria, and Romania [2]. One of the main characteristics of BEN is the frequent development of urinary tract tumors, occurring in about 40% of BEN patients [3]. There have been many different hypotheses on BEN pathophysiology, among which those including exposure to ochratoxin (OTA) and aristolochic acid (AA) are the most discussed ones. Both OTA and AA are toxic compounds with different chemical structure and origin. While OTA is a mycotoxin produced by fungus belonging to *Aspergillus* and *Penicillium genera* [4,5], AA is synthesized by *Aristolochia clematitis*, a weed growing in wheat fields in the Balkan region [6,7]. Ochratoxin can contaminate a large number of food commodities and food products [8,9], while AA is frequently found in cereals and cereal products, as well as in Chinese medicinal products [10,11,12,13].Very recently, Stoev et al. addressed the exposure levels of humans to OTA and the possible synergism or additive effects between OTA and some target mycotoxins such as citrinin, penicillic acid and fumonisin B1. All these mycotoxins are often found in the Balkans as contaminants of feedstuffs/foods from endemic areas [14].

It is important to note that cellular metabolism of both major nosologic entities of BEN, OTA, and AA includes complex metabolic activation, associated with the production of reactive oxygen species, resulting in oxidative distress [3]. In the course of OTA metabolism in microsomes, in the presence of NADPH and O_2_, hydroxyl radicals are produced even without exogenous Fe, followed by increased levels of oxidative stress byproducts [15]. Besides, enhanced intracellular accumulation of free radicals also might be a secondary event, as the consequence of mitochondrial alterations induced by OTA [16]. Extracellular release of byproducts of oxidative damage results in systemic oxidative stress documented in humans and animal models of OTA toxicity, as reviewed in detail in a recent study of Marin-Kuan et al. [17]. This kind of OTA-mediated redox imbalance in animals may be ameliorated by the administration of recombinant mitochondrial antioxidative enzyme superoxide dismutase (SOD2), responsible for dismutation of superoxide into the hydrogen peroxide [18]. Moreover, treatment with recombinant SOD2 (rSOD2) can diminish not only oxidative distress induced in rats but also OTA-induced renal proximal tubular damage [18]. Regarding AA metabolism, few enzymes and coenzymes are implicated in AA bioactivation, including hepatic cytochrome P450 enzymes (CYP1A1, CYP1A2) and NAD(P)H: quinone reductase (NQO1), as well as renal NQO1, cytochrome P450 reductase (CPR), cyclooxygenase (COX), and sulphotransferase (SULT1A1). These enzymes are involved in biotransformation of AA to N-hydroxyaristolactam I, which can be oxidized by renal peroxidases to form persistent adducts with DNA. These adducts are known to be highly mutagenic and can initiate tumors [19], whereas AA-induced oxidative stress may play an important role in the development of renal injury in BEN [20].

The involvement of antioxidant and detoxification enzymes, glutathione transferases (GST) has been recently suggested in BEN patients [21,22]. Namely, it has been shown that carriers of variant genetic allele, encoding GSTA1 enzyme associated with lower GSTA1 expression, have a higher risk of BEN [23]. Since GSTA1 is capable of inactivating lipid hydroperoxides, it seems reasonable to speculate that altered antioxidative defense can also contribute to oxidative stress in BEN patients. It should be emphasized that *SOD2* gene is also polymorphic and (6q25) has several SNPs, among which rs4880 C > T is the most studied. The SNP in the *SOD2* gene causes an amino acid substitution of alanine (*Ala)* with valine (*Val*) (*Ala16Val*) [24]. *Val* allele of this SNP results in structural alterations in the mitochondrial targeting domain of SOD2, leading to its less efficient post-transcriptional transport into the mitochondrion and decreased potential in neutralizing superoxide anion [24,25]. Individuals with the *CC* (*Ala16Ala*) genotype were found to have higher SOD2 activity compared to those with the *CT* (*Ala16Val*) or *TT (Val16Val*) genotype [26]. Superoxide dismutase works in a mutually supportive fashion with another selenium-dependent antioxidant enzyme glutathione peroxidase (GPX). Glutathione peroxidases represent a family of enzymes catalyzing the reduction of hydrogen peroxide to water [27]. GPX1 is the most abundant and ubiquitous isoform. One of the most extensively studied *GPX1* polymorphisms represents modified C > T (GPX1C593T, dbSNP ID rs1050450), which leads to a change from proline (*Pro*) to leucine (*Leu*) at amino acid 200 (*Pro200Leu*) [27]. The *GPX1* C593T variant is supposed to result in lower enzyme activity and mRNA expression in the presence of *Leu*-allele compared with the *Pro*-allele [28]. It is important to note that polymorphic expression of either *SOD2* and/or *GPX1* influences susceptibility to end-stage renal disease (ESRD) and transitional cell carcinoma (TCC), both associated with BEN progression and complications [29,30,31].

Based on experimental data on the involvement of oxidative distress in the metabolism of both potential causes of BEN and clinical data on the role of *SOD2* rs4880 or *GPX1* rs1050450 polymorphisms in ESRD and TCC risk as major BEN complications, we hypothesized that these genetic polymorphisms also influence the risk of BEN and its associated tumors. In order to assess the role of *SOD2* rs4880 and *GPX1* rs1050450 polymorphisms as risk determinants of BEN and BEN associated tumors, we conducted a genetic study in 207 patients with verified presence of Balkan endemic nephropathy, recruited from three hemodialysis centers in Republic of Serbia and Bosnia and Herzegovina, and respective 86 healthy controls from endemic areas.

## 2. Materials and Methods

### 2.1. Study Participants

A total of 293 participants (207 patients with Balkan endemic nephropathy and 86 healthy controls) were enrolled in this study. BEN was diagnosed using previously defined criteria [32]. All BEN patients were recruited from three hemodialysis centers (Bijeljina and Šamac from Bosnia and Herzegovina—Republic of Srpska and Lazarevac from the Republic of Serbia). The control group was comprised of residents of endemic areas with: Negative family history of BEN, verified absence of endemic nephropathy by clinical, laboratory, and echosonographical examination, verified absence of other renal diseases, hypertension, diabetes, and malignant tumors. An informed consent form was obtained from all participants and the research was carried out in compliance with the Helsinki Declaration. This study was approved by the Ethical Committee of the Faculty of Medicine, University of Belgrade (permission number 29/VI-13, 25 June 2012).

### 2.2. DNA Isolation

A total DNA was purified from whole blood using the QIAmp^®^ DNA Blood Mini kit (Qiagen, Inc., Chatsworth, CA, USA). Concisely, after lysing the cell membranes of leukocytes in detergent solution, all proteins linked to DNA underwent enzymatic digestion by proteinase K. The lysates were transferred into mini spin columns with silica membranes, selective for DNA binding. DNA was obtained after a series of washing steps with buffers which contain salts and ethanol. Isolated DNA was stored at −20 °C until polymerase chain reaction (PCR) was performed.

### 2.3. Analysis of SOD2 rs4880 and GPX1 rs1050450 Genotypes

*SOD2* (rs4880) and *GPX1* (rs1050450) polymorphisms were determined by the real-time PCR (qPCR). The method was performed on Mastercycler® ep realplex (Eppendorf, Hamburg, Germany), using TaqMan Drug Metabolism Genotyping assay *SOD2* (Applied Biosystems, ID: C_8709053_10) and *GPX1* (Applied Biosystems, ID: C_1623856). PCR reaction mixture for each sample of the assessed polymorphisms contained: 5 μL DNA, 0.125 μL TaqMan probe, 2.5 μL PCR master mix, and 2.375 μL water. The amplification reaction of the DNA segment consisted of 30 repeated cycles through three steps of denaturation (4 min at 94 °C), annealing (30 s at 60 °C) and extension (45 s 72 °C), followed by final elongation of 5 minutes at 72 °C. Results of genotyping were visualized by Mastercycler® ep realplex software (Eppendorf, Hamburg, Germany).

### 2.4. Statistical Analysis

The Statistical Package for the Social Sciences 17 (SPSS Inc., Chicago, IL, USA) was used to calculate the significance of the studied variables. Student’s t-test was used to compare continuous variables (age, BMI), after confirming normality of the distribution and the equality of variances by Kolmogorov–Smirnov and Levene’s tests. Differences between the groups for categorical variables (gender, smoking status, transitional cell carcinoma, *SOD2* and *GPX1* genotypes) were compared using χ2 test. This test was also used in order to test deviation of the genotype distribution from Hardy–Weinberg equilibrium. Multinomial logistic regression was applied to assess the role of SOD2 and GPX1 gene polymorphism on BEN risk, including age and gender as the possible confounders. A *p*-value less than 0.05 was considered statistically significant.

## 3. Results

A total of 207 patients with BEN (44% female, 56% male) and 86 healthy controls from BEN area (56% female and 44% male) participated in this study. Table 1 shows selected characteristics of BEN patients and healthy controls. As expected, patients and controls differ in terms of body mass index (BMI). Namely, BEN patient had lower BMI when compared to controls (24.61 ± 4.36 vs. 26.2 ± 3.71, *p* = 0.004). In our study, patients were a bit older than the control group (70.68 ± 6.6 vs. 74.68 ± 6.66 (*p* ˂ 0.001). We found that a total of 30 (14%) BEN patients developed transitional cell carcinoma.

The distribution of *SOD2* (rs4880, C > T) and *GPX1* (rs1050450, C > T) polymorphisms among BEN patients and controls is presented in Table 2. All genotype frequencies were in Hardy-Weinberg equilibrium for cases and controls. Our results showed that variant *SOD2 Val/Val* genotype was more frequent in patients with BEN (27%) than in the control group (14%). Moreover, individuals carrying this *SOD2 Val/Val* variant homozygous genotype were at a 2.6-fold higher risk of BEN development (OR = 2.6, 95% CI: 1.15–5.85, *p* = 0.021) when compared to individuals with referent *SOD2 Ala/Ala* genotype. No significant difference was observed in the distributions of examined *GPX1* rs1050450 polymorphism, and there was no association of different *GPX1* genotypes with BEN risk independently.

The association between the combined effect of *GPX1* and *SOD2* genotypes and the risk of Balkan endemic nephropathy is shown in Table 3. The results indicated that there was no significant effect between polymorphism of these two genes and BEN risk. However, a certain trend of decrease in risk was observed for individuals with a combination of wild type *SOD2* genotype and at least one variant *GPX1* allele (*Ala/Ala / Pro/Leu+Leu/Leu)*, although still not statistically significant (OR = 0.45, 95% CI: 0.16–1.25, *p* < 0.125).

Since the incidence of TCC of the urinary tract in BEN patients is much higher than in general population, we also analyzed the association between *SOD2* rs4880 and *GPX1* rs1050450 gene polymorphisms and TCC in patients with BEN (Table 4). The results of this case-only analysis showed that patients with at least one *SOD2 × Val* allele had increased risk of TCC up to 1.55 (95% CI: 0.52–4.68), however without reaching statistical significance (*p* = 0.433). On the other hand, regarding *GPX1* polymorphism, patients with at least one copy of variant *GPX1* allele exhibited decreased susceptibility towards TCC development in comparison to *GPX1* wild type homozygotes, which, however, was not statistically significant either (OR = 0.57, 95% CI: 0.26–1.28, *p* = 0.175).

## 4. Discussion

The results of this study have shown that individuals with both copies of variant *SOD2* allele, known to be associated with lower mitochondrial antioxidant protection, are at a significantly higher risk of Balkan endemic nephropathy. Polymorphic expression of *GPX1* was not significantly associated with the risk of Balkan endemic nephropathy, either individually or in combination with *SOD2* variant genotype. Regarding susceptibility to TCC in BEN patients, none of the polymorphisms studied was significantly associated with the risk of TCC. However, the presence of at least one copy of variant *SOD2* allele was associated with a certain increase in risk, while the presence of at least one variant *GPX1* allele was associated with a modest decrease in risk, when compared to BEN patients with both referent alleles of the corresponding genes.

Some dietary products or/and drinks contain numerous mycotoxins in small amounts, but periodic exposure to them for a prolonged period may become hazardous for humans or animals [33]. Recent data suggest there is a joint mycotoxin interaction in complex etiology of BEN. Moreover, there is some evidence that OTA might enhance AA-induced uroepithelial malignancy in BEN [14]. Since these processes might be mediated by free radicals produced in the course of OTA or/and AA metabolism, it was speculated the variant *SOD2* allele would be associated with a higher risk of Balkan endemic nephropathy. Indeed, our data demonstrated that carriers of one copy of variant *SOD2* allele in areas of Balkan endemic nephropathy had higher risk of BEN, when compared to individuals with both wild type *SOD2* gene variants, although the difference observed was not statistically significant. However, individuals with both copies of variant *SOD2* were at 2.5 times higher risk of BEN when compared to corresponding controls, originating from BEN areas.

Namely, effective antioxidant protection against superoxide anion, provided by adequate SOD2 enzyme availability, prevents mycotoxin or AA-induced cellular damage, with special emphasis on mitochondrial compartment. In BEN, chronic interstitial disease-specific morphological changes, such as tubular atrophy and/or mild vascular changes occur. Savin et al. suggested that nephrotoxins, such as ochratoxin, may affect tubular cell function, which triggers apoptosis of tubular cells [34]. This is in agreement with the study of Bouaziz et al. who showed that exposure of Hela cells to ochratoxin induces apoptosis by specific alterations of mitochondria in addition to enhanced superoxide production and release of cytochrome C [16]. Regarding the role of mitochondria in OTA metabolism in human kidney cells, common characteristic for all types of ochratoxicosis is impaired energy production and disorder in membrane structure which results in the loss of membrane stability of mitochondria and other cell organelles in the of proximal renal tubule cells. Indeed, rat liver mitochondria in vitro studies showed that OTA inhibited the respiration in mitochondria acting as a competitive inhibitor of its inner membrane carrier proteins [35,36]. Moreover, in that manner, OTA also inhibited phosphate transport into mitochondria, resulting in disruption of the mitochondria [35,36]. It might be speculated that in the course of chronic OTA exposure of individuals living in BEN areas, overall cellular production of free radicals increases, with special emphasis on the susceptible kidney structures, such as tubules and their mitochondria, thus initiating mitochondrial cell death. Such scenario, that underlies the central role of mitochondrial ROS in OTA-induced programmed cell death, has recently been documented in *Arabidopsis thaliana*, by means of mitoproteome analysis [37]. Namely, in this study, it was demonstrated that ochratoxin was able to increase the production of mitochondrial ROS and influence change in mitochondrial membrane potential, as well as the release of cytochrome C into the cytosol which eventually led to apoptosis [37]. Moreover, results of this study about digital gene expression data at the transcriptional level indicated that mitochondrial dysfunction was a prerequisite for OTA-induced programmed cell death and the initiation and execution of programmed cell death via a mitochondrial-mediated pathway. Further studies are needed to elucidate whether this pathway is also activated after OTA exposure in human tubulocytes and animal models of chronic OTA exposure. 

In addition to OTA, AA impaired redox hemostasis, which has been clearly demonstrated in several recent studies suggests shared mechanisms for nephrotoxicity. Recent comparative analysis of the effects of OTA and AA on biomarkers of antioxidant activity in piglets showed that AA decreased the activity of catalase (CAT), superoxide dismutase (SOD), and glutathione peroxidase (GPx) in the liver and kidneys, while OTA decreased only GPx activity in the kidneys. It seems that altered antioxidant enzyme activity, which in the case of this study is affected by gene polymorphism of these enzymes, may lead to oxidative damage to kidneys. Indeed, our data on higher susceptibility to BEN in homozygous carriers of *SOD2* variant allele confirm this thesis. Interestingly, the pro-oxidative effect of AA was preventable by vitamin E in rat renal tubular cells. Vitamin E (α-tocopherol) ameliorates aristolochic acid-induced renal tubular epithelial cell death by attenuating oxidative stress and caspase-3 activation [38] pointing out that not only combination of mycotoxins or alkaloids such as AA play a role in nephrotoxicity, but also dietary intake of antioxidant vitamins interfere with the risk. It seems that impaired redox switch Nrf2 system, and profound changes in lipid metabolites are involved too [39]. Although antioxidant enzymes in our study do not belong to Nrf2 targets, their response might be important in prolonged inflammatory and pro-oxidative challenge in the course of combined mycotoxin and AA exposure. Namely, SOD2 belongs to Nf-kappa target which, together with Nrf2, interlinks oxidative stress and inflammation in BEN [40].

Interestingly, genetic polymorphism rs1050450 in *GPX1*, a major antioxidant enzyme responsible for hydrogen peroxide inactivation, does not seem to influence susceptibility to Balkan endemic nephropathy. Moreover, variant *GPX1* allele in combination with protective wild type *SOD2* genotype conferred twice decreased risk (OR = 0.45, 95% CI: 0.16–1.25), although it was not statistically significant. Still, this trend is worth mentioning, since there is literature data on better overall survival in carriers of variant low-activity *GPX1* (*Leu*) allele [41]. In the view of the fact that variant *GPX1* exhibits lower enzyme activity, the explanation of such phenomenon is challenging. As recently suggested, the roles of hydrogen peroxide in signal transduction and regulation of genes involved in longevity might have priority when compared to its potential to cause oxidative damage [42]. Further studies are needed to elucidate mechanisms by which altered H_2_O_2_ reduction is associated with better survival and lower susceptibility to BEN.

Regarding the influence of polymorphisms in genes encoding antioxidant enzymes and risk of transitional cell carcinoma, we showed that the presence of variant *SOD2* allele modifies the cancer risk in BEN patients by a 1.5-fold increase. This result is in accordance with the study of Hung et al. who have also shown that patients with *SOD2 Val/Val* genotype had an increased risk of bladder cancer [31]. However, recent meta-analyses resulted in quite inconsistent data regarding *SOD2* rs4880 and *GPX1* rs1050450 polymorphisms. Thus, meta-analysis on 15,320 cancer cases and 19,534 controls from 34 published case-control studies demonstrated no significant overall main effect of *SOD2* rs4880 polymorphism on cancer risk [6]. Besides, the latest meta-nalysis of Cao et al. showed that the *SOD2* rs4880 polymorphism was not associated with bladder cancer risk, while *GPX1* rs1050450 polymorphism significantly increased susceptibility to bladder cancer [43]. 

There are certain limitations of this study that should be considered. The first one is the relatively small sample size that is in favor of creating potential biases. The second is the fact that the control group consisting of elderly-focused healthy individuals from BEN region was relatively small. Furthermore, apart from age and gender, more confounding factors might have influenced the results.

## 5. Conclusions

The results of this study provide some important novel aspects regarding the role of mitochondrial *SOD2* rs4880 gene polymorphism in pathophysiology of BEN. Given its previously described roles, we can hypothesize that polymorphic expression of SOD2 affects superoxide accumulation in ochratoxin-induced mitochondrial distress and induction of programmed cell death in human kidneys contributing to renal damage and development of Balkan endemic nephropathy. Therefore, further investigations with larger sample size and more rigorous designs and mechanistic studies are needed to elucidate the role of antioxidant enzymes *SOD2* rs4880 and *GPX1* rs1050450 polymorphisms in modifying the BEN risk, as well as, the risk of uroepithelial tumors in BEN patients. Besides, in the omics era, SOD2 genotyping could, together with a panel of other enzymes shown to be associated with susceptibility to BEN, (*GSTA1, GSTM1, GSTT1, *CYP3A5, NQO1**) be potentially applied as a biomarker of susceptibility in BEN areas [23,44,45,46]. 

## Figures and Tables

**Table 1 medicina-55-00435-t001:** Demographic characteristics of patients with Balkan endemic nephropathy and controls.

Characteristic	Controls	Patients	*p*
Age (years)	74.68 ± 6.66	70.68 ± 6.60	<0.001
Gender, *n* (%)			
Female	48 (56)	90 (44)	
Male	38 (44)	117 (56)	0.054
Smoking, *n* (%)			
Never	64 (77)	137 (66)	
Ever	19 (23)	70 (34)	0.513
BMI	26.20 ± 3.71	24.61 ± 4.36	0.004
Transitional cell carcinoma *n* (%)			
No	/	177 (86)	
Yes	/	30 (14)	

All results are presented as mean ± SD or frequencies.

**Table 2 medicina-55-00435-t002:** Association of *SOD2* and *GPX1* genotypes with the risk of Balkan endemic nephropathy.

Genotype	Controls, *n* (%)	Patients, *n* (%)	OR (95% CI)	*p*
*SOD2* rs4880				
*Ala/Ala*	28 (33)	43 (22)	1.00 ^a^	
*Ala/Val*	45 (53)	99 (51)	1.34 (0.72–2.49)	0.352
*Val/Val*	12 (14)	52 (27)	2.6 (1.15–5.85)	0.021
*Ala/Val* + *Val/Val*	57 (67)	151 (78)	1.61 (0.89–2.91)	0.111
*GPX1* rs1050450				
*Pro/Pro*	42 (49)	87 (43)	1.00 ^a^	
*Pro/Leu*	35 (41)	92 (46)	1.25 (0.72–2.18)	0.436
*Leu/Leu*	9 (10)	22 (11)	1.06 (0.44–2.58)	0.9
*Pro/Leu* +*Leu/Leu*	44 (51)	114 (57)	1.21 (0.71–2.05)	0.483

^a^ Reference category. OR, odds ratio; CI, confidence interval; Adjustments: Age, gender.

**Table 3 medicina-55-00435-t003:** Association between combined *GPX1* and *SOD2* genotypes and the risk of Balkan endemic nephropathy.

Combined *SOD2/GPX1*	Controls *n* (%)	Patients *n* (%)	OR (95% CI)	*p*
*Ala/Ala / Pro/Pro*	13 (15)	23 (12)	1.00 ^a^	
*Ala/Ala / Pro/Leu+Leu/Leu*	15 (18)	17 (9)	0.45 (0.16–1.25)	0.125
*Ala/Val+Val/Val / Pro/Pro*	29 (34)	60 (32)	0.9 (0.38–2.12)	0.811
*Ala/Val+Val/Val / Pro/Leu+Leu/Leu*	28 (33)	88 (47)	1.41 (0.6–3.29)	0.432

^a^ Reference category. OR, odds ratio; CI, confidence interval; Adjustments – age, gender.

**Table 4 medicina-55-00435-t004:** Association of *SOD2* and *GPX1* polymorphisms with transitional cell carcinoma in patients with Balkan endemic nephropathy.

Transitional Cell Carcinoma				
Genotype	No *n* (%)	Yes *n* (%)	OR (CI 95%)	*p*
*SOD2* rs4880				
*Ala/Ala*	38 (23)	5 (17)	1.00 ^a^	
*Ala/Val*	82 (50)	17 (59)	1.55 (0,52–4,68)	0.433
*Val/Val*	45 (27)	7 (24)	1.39 (0,39–4,93)	0.611
*Ala/Val*+*Val/Val*	127 (77)	24 (83)	1.48 (0,52–4,27)	0.465
*GPX1* rs1050450				
*Pro/Pro*	71 (42)	16 (53)	1.00 ^a^	
*Pro/Leu*	81 (47)	11 (37)	0.54 (0,23–1,27)	0.161
*Leu/Leu*	19 (11)	3 (10)	0.72 (0,18–2,85)	0.644
*Pro/Leu*+*Leu/Leu*	100 (58)	14 (47)	0.57 (0,26–1,28)	0.175

^a^ Reference category. OR, odds ratio; CI, confidence interval; Adjustments – age, gender.

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
