# Peer review of "Association of SOD2 (rs4880) and GPX1 (rs1050450) Gene Polymorphisms with Risk of Balkan Endemic Nephropathy and its Related Tumors"

_medicina, 2019, doi:10.3390/medicina55080435_

Round 1

Reviewer 1 Report

Dragicevic and colleagues have performed an interesting study about the association of two polymorphisms (SOD2 rs4880 and GPX1 rs1050450) with BEN (Balkan Endemic Nephropathy) frequency. They found that rs4880 seems to affect the risk of BEN development, while no association was detected between rs1050450 and disease.

The paper is well written, interesting and experiments well performed. According to this reviewer only minor issues need to be addressed.

According to SNP database the rs4880 is a Val16Ala c.47T>C. However, the authors consider Ala as “most frequent allele” and refer to this SNP as Ala16Val C>T. Is this due to the fact that the population here investigated is presumably from balcanic region and then Ala is the most frequent allele? If yes, this should be explained

Please indicate the SNPs in univocal way through the text. They could be described the first time they are reported and then the authors can refer to them with SOD2 rs4880 and GPX rs1050450

Introduction (line 68): please indicate what rSOD2 means (probably recombinant SOD?)

Results (line 190): what the authors should explicitate what they mean with “SOD2*Val allele”

Results section. The authors assert that smoking prevalence among BEN patients was higher than in controls. However, this difference is not statistically significant (Table 1). This data is quite useless without statistical significance, or at least it should be shortly discussed.

All gene should be italicized through the text

Author Response

We thank the reviewer for nice words. We accepted all suggestions and corrections.

Point 1: According to SNP database the rs4880 is a Val16Ala c.47T>C. However, the authors consider Ala as “most frequent allele” and refer to this SNP as Ala16Val C>T. Is this due to the fact that the population here investigated is presumably from balcanic region and then Ala is the most frequent allele? If yes, this should be explained

Response 1: We thank the reviewer for this comment. Indeed we refer SNP SOD2 rs4880 as Ala16Val C>T. The reason for that is the fact that in our control group Ala is the most frequent allele. Namely, SOD2 Ala16Ala genotype frequency is 33%, while SOD2 Val16Val genotype is 14% in our controls. The majority of Caucasian population has a similar frequency of SOD2 rs4880 genotypes (Craword et al., 2012)

Point 2Please indicate the SNPs in univocal way through the text. They could be described the first time they are reported and then the authors can refer to them with SOD2 rs4880 and GPX rs1050450

Response 2: We accepted this suggestion and SNPs are changed to be in univocal way through the text.

Point 3:  Introduction (line 68): please indicate what rSOD2 means (probably recombinant SOD?)

Response 3: Yes, rSOD2 means recombinant SOD2. Therefore we added “recombinant SOD2” in Introduction section (line 68).

Point 4:     Results (line 190): what the authors should explicitate what they mean with “SOD2*Val allele”

Response 4: We mean that individuals carriers of at least one SOD2*Val allele had increased risk of TCC up to 1.55. Therefore we added ”at least one” in the text of Results section (line 189).

Point 5:    Results section. The authors assert that smoking prevalence among BEN patients was higher than in controls. However, this difference is not statistically significant (Table 1). This data is quite useless without statistical significance, or at least it should be shortly discussed.

      Response 5: We completely agree with the reviewer. Therefore the sentence “Interenstigely,  the smoking prevalence among BEN cases was higher (34%) than in controls (23%).” has been deleted from Results section.

Point 6:  All gene should be italicized through the text

Response 6: We checked all genes and corrected formatting.

Reviewer 2 Report

Transitional cell carcinoma is rare to kidney cancer. Authors should also analyze the association between the SOD2 polymorphism and other types of kidney cancer, especially for RCC.

Delete o in line 216 of page 6.

Author Response

We thank the reviewer.

Point 1:      Transitional cell carcinoma is rare to kidney cancer. Authors should also analyze the association between the SOD2 polymorphism and other types of kidney cancer, especially for RCC.

Response 1: In our study we investigated the association between SOD2 rs4880 and GPX1 rs1050450 SNPs and urinary tract transitional cell carcinoma in patients with BEN. Namely, it is well known that BEN is associated with increased incidence of urinary tract transitional  cell carcinoma which is localized, either in renal pelvis and ureter, or in renal pelvis and urinary bladder (Stefanovic and Cosyns, 2004). We did not analyze the association between the SOD2 rs4880 polymorphism and renal cell carcinoma (RCC), since BEN is not associated with this type of kidney cancer.

            Point 2:. Delete o in line 216 of page 6.

Response 2: We have deleted o in line 216 of page 6.

Reviewer 3 Report

The authors indicate that “Association of SOD2 (rs4880) and GPX1 (rs1050450) Gene Polymorphisms with Risk of Balkan Endemic Nephropathy and its Related Tumors”. Due to limitations in environmental and health science elements of exposure to OTA, could not be linked to BEN as causative factor. I have some following comments for authors need to be addressed.

1.      OTA have gender biasness in its toxicological effects, does this factor was considered while enrolling the patients?

2.      Any biomarkers which can clearly established the mechanism by which OTA causes BEN? I mean the level of OTA and AA checked in blood or urine from BEN patients and non BEN patients.

3.      Experiments to show the oxidative stress data and estimation of GST is missing? Needs to be justified.

4.      In discussion part line#216, it was --- o speculated?

5.      The author mentioned in discussion part line # 217, that the carriers of one copy of variant SOD2 allele in areas of Balkan endemic nephropathy had higher risk of BEN when compared to individuals with both wild type SOD2 gene variants, however percentage of one copy of variant SOD2 allele in controls are more compared to patients?

6.      They need to estimate apoptosis assays also to confirm the OTA effects to see tubular and interstitial damages in BEN.

7.      In discussion part line 242, the author said that in this study, it was demonstrated OTA was able to increase the production of mitochondrial ROS, changes in mitochondrial potential and also release of cytochrome C, eventually leads to apoptosis (37). The author put a reference, needs to explain. There is no experimental data.

Author Response

            We thank the reviewer for her/his usefull comments.

            Point 1: The authors indicate that “Association of SOD2 (rs4880) and GPX1 (rs1050450)                 Gene Polymorphisms with Risk of Balkan Endemic Nephropathy and its Related                  Tumors”.Due to limitations in environmental and health science elements of exposure to OTA, could             not be linked to BEN as causative factor. I have some following comments for authors need to  be addressed.

Response 1: We completely agree that OTA could not be link to BEN as causative factor.

Point 2. OTA have gender biasness in its toxicological effects, does this factor was considered while enrolling the patients?

     Response 2: For all associations data were adjusted according to gender. Gender distribution  did not significantly differ between BEN patients and controls.

Point 3: Any biomarkers which can clearly established the mechanism by which OTA causes BEN? I mean the level of OTA and AA checked in blood or urine from BEN patients and non BEN patients.

Response 3: We thank reviewer for her/his useful comment. We were not able to perform sophisticated measurements of OTA and AA metabolites in blood and urine samples of BEN patients. The main goal of our study was to assess whether susceptibility to BEN is modified by antioxidant enzymes polymorphisms. We believe that future studies addressing in-depth analyzes of toxicological phenotype e.g. level of OTA and AA checked in biological samples in context of polymorphic expression of both antioxidant and detoxification enzymes will provide better insight into the BEN pathogenesis.

Point 4:  Experiments to show the oxidative stress data and estimation of GST is missing? Needs to be justified.

Response 4: There is available evidence that the levels of oxidative byproducts are increased. In the future studies we plan to analyze oxidative stress byproducts in relation to genotype according to the design we used in our previous paper (Suvakov et al., 2013; Suvakov et al., 2019)

            Point 5:In discussion part line#216, it was --- o speculated?

Response 5: We have deleted o in line 216 of page 6.

        Point 6: The author mentioned in discussion part line # 217, that the carriers of one copy of             variant SOD2 allele in areas of Balkan endemic nephropathy had higher risk of BEN when                 compared to individuals with both wild type SOD2 gene variants, however percentage of                 one copy of variant SOD2 allele in controls are more compared to patients?

         Response 6:We thank the reviewer for this comment. However, the percentage of one copy of variant  SOD2 allele in controls was not higher compared to patients. Namely, frequency of            variant  SOD2  Val/Val in controls was 14% while this SOD2 genotype was present in 27%                of BEN patients.

 Point 7:   They need to estimate apoptosis assays also to confirm the OTA effects to see tubular and interstitial damages in BEN.

Response 7: There are some literature data that OTA could induce the apoptosis of human tubular kidney cells (HKC) in-vitro, with the  possible  mechanism  of  activating  JNK  and  then upregulating the protein expression of caspase-3 (Li et al., 2012). However, to the best of our knowledge specific OTA induced changes have not been confirmed in pathological samples, in biopsies, from BEN patients.

Point 8:    In discussion part line 242, the author said that in this study, it was demonstrated OTA was able to increase the production of mitochondrial ROS, changes in mitochondrial potential and also release of cytochrome C, eventually leads to apoptosis (37). The author put a reference, needs to explain. There is no experimental data.

Response 8: We thank the reviewer for this comment. To additional clarify this statement we introduce the following text describing data from reference 37 “Moreover, results of this study about digital gene expression data at transcriptional level indicated that mitochondrial dysfunction was a prerequisite for OTA-induced programmed cell death and the initiation and execution of programmed cell death via a mitochondrial-mediated pathway.”(lines 244-247).